

# Substorm Signatures in the Dayside Magnetosphere

Sanjay Kumar [1] and Tuija I. Pulkkinen [1]

[1]Department of Climate and Space Sciences and Engineering, University of Michigan, Ann Arbor, MI, USA

**Correspondence:** Sanjay Kumar (kumarsa@umich.edu)

**Abstract.** We investigate variations in the position of the magnetopause in response to the interplanetary magnetic field (IMF), and different phases of magnetospheric substorms. The average location of magnetopause is examined using magnetic field observations from multiple satellites (THEMIS, RBSP, and MMS), and the Shue model utilizing OMNI solar wind data for a period of five years from 2016-2020. We estimate average position of the magnetopause using Shue model through superposed

epoch analysis of standoff distance and tail flaring angle at different substorm timings (onset, peak and end) and from in-situ measurements through 2D equatorial maps of average $\Delta B_Z$ under IMF $|B_z| > 0$ conditions. Our findings reveal the occurrence of substorms during both northward and southward IMF orientations and highlight an earthward movement of the magnetopause during substorm onset and peak, followed by a relaxation during the substorm end time, for both northward and southward IMF orientations. Notably, the magnetopause undergoes significant compression and reaches its closest point

to the Earth during instances of strong southward IMF ($B_Z < -5$), particularly during the substorm peak. The empirical model provides accurate estimation of the magnetopause location during periods of both strong northward and southward IMF $|B_z| > 5$, as the model curve traverses a distinct location ($\Delta B_Z = 0$) representing the magnetopause shown in the 2D average map of $\Delta B_Z$.

## 1  Introduction

The magnetopause is the boundary of the Earth's magnetosphere which separates the magnetic cavity from the surrounding plasma environment. The location of the magnetopause is determined by the pressure balance between magnetospheric magnetic field and the solar wind. The magnetopause boundary is not stationary, being strongly influenced by the solar wind dynamic pressure (Chapman and Ferraro, 1931), the interplanetary magnetic field (IMF) orientation and strength (Fairfield, 1971; Shue et al., 1997, 1998), and dipole tilt angle (Liu et al., 2012). The solar wind pressure changes move the magnetopause

boundary, sometimes to inside geosynchronous orbit ($\sim 6.6\ R_E$, $R_E$ = Earth radius) (Cahill and Winckler, 1999). Furthermore, strongly southward IMF leads to inward motion of the magnetopause due to magnetic flux erosion from the dayside magnetopause via magnetic reconnection (Dungey, 1961).

Several models parameterize the magnetopause location and shape by solar wind and IMF parameters (Chao et al., 2002; Fairfield, 1971; Sibeck et al., 1991; Lin et al., 2010; Liu et al., 2015; Nguyen et al., 2022, and references therein). Shue et al.

(2000) reviewed many magnetopause models and compared the differences among them for extreme solar wind conditions and their limitations. Shue et al. (1997) studied the magnetopause location using in-situ magnetopause crossings by multiple





satellites to construct an empirical model that incorporates the influence of solar wind dynamic pressure and IMF $B_Z$ on controlling the location and shape of the magnetopause.

Wang et al. (2018) studied the effects of IMF north-south orientation and upstream solar wind dynamic pressure on the location of the magnetopause and bow shock using a global MHD model. They found that during southward IMF and high solar wind pressure, increased reconnection moves the magnetopause earthward and outward for positive IMF $B_Z$. They also conclude that the effect of dynamic pressure on magnetopause location is more prominent than those due to the IMF orientation changes. Lu et al. (2011) constructed a magnetopause model through global MHD calculations and observed that IMF $B_Z$ primarily influences the magnetopause shape with minor effects on standoff distance. In contrast, solar wind dynamic pressure predominantly affects the magnetopause standoff distance with minimal impact on the magnetopause shape.

Substorms are transient phenomena that occur in the Earth's magnetotail, storing and releasing solar wind energy through an explosive process (Baker et al., 1996). Substorms represent a key dynamic cycle in the solar wind – magnetosphere – ionosphere system, with the coupling involving intensification of auroral currents (Akasofu, 1964). Several studies have proposed that substorms are triggered by changes in the solar wind driver: While substorm onsets are often followed by an interval of southward IMF (Kokubun, 1972), northward turnings of the IMF can also be responsible for triggering substorms (Mcpherron et al., 1986; Sergeev et al., 1986). While Wild et al. (2009) concluded that substorm onsets occur following an IMF southward turning and at least 20-min interval of southward IMF. Furthermore, Hsu (2003) considered changes in IMF $B_Y$, dynamic pressure, and IMF $B_Z$ changes and concluded that majority of the substorms are triggered by IMF $B_Z$ change, while a rather small number are triggered by IMF $B_Y$ rotation or change of dynamic pressure, while some substorms have no identifiable external trigger (Henderson et al., 1996). Aubry et al. (1970) observed inward motion of magnetopause and its relation to an increase in the tail flux and substorm onset using satellite observations. They found earthward motion of magnetopause during reversal of IMF $B_Z$ from northward to southward just prior to substorm onset which continues for two hours with the magnetopause moving inward up to 2 $R_E$.

In this paper we present statistical investigation of average location of magnetopause boundary for strong northward-southward IMF during different phases of substorms. Focusing on a period of 5 years from 2016-2020, we use satellite observations from Radiation Belt Storm Probes (RBSP) (Mauk et al., 2013), Time History of Events and Macroscale Interactions during Substorms (THEMIS) (Angelopoulos, 2008), and Magnetospheric Multiscale (MMS) (Burch et al., 2016), which provide a very good coverage of magnetosphere out to 30 $R_E$ in the dayside. We complement the space measurements with data from ground-based magnetometers available from the SuperMAG collaboration (Gjerloev, 2012). For this study period we identified 5077 substorms from a list of substorm onsets created by Ohtani and Gjerloev (2020). We use superposed epoch analysis to estimate the average standoff distance and tail flaring angle taken from the nonlinear relation given by Shue et al. (1998) in their empirical model for magnetospheric shape and size. We also discuss the application of Shue model in the estimation of average magnetopause location. Section 2 describes the data, Section 3 presents average map of observed magnetic field in the equatorial plane during substorms phases and Section 4 presents a superposed epoch analysis, 5 shows the empirical model by Shue and Section 6 concludes with discussion of results.



## 2 Data

We examine the magnetospheric signatures of substorms during the interval of 2016–2020, when several (multisatellite) missions were operational. We use data from the three Time History of Events and Macroscale Structures during Substorms (THEMIS) in near-Earth near-equatorial orbits, from the two Radiation Belt Storm Probes (RBSP) in the inner magnetosphere inside of about $\sim 6R_E$, and from one of the Magnetospheric Multiscale (MMS) spacecraft in near-equatorial, higher-altitude orbit. Although the MMS mission involves four spacecraft, their close formation is such that incorporating observations from more than one spacecraft is not pertinent to this study.

We use magnetic field data from the EMFISIS instrument suite (Kletzing et al., 2013) onboard both RBSP-A and RBSP-B spacecraft. We also use spin-averaged magnetic field data from the Fluxgate Magnetometer (FGM) (Auster et al., 2008) from THEMIS-A,D, and E (Excluding THEMIS-B and THEMIS-C, which orbit around the Moon). Magnetic field data from the MMS-1 spacecraft come from the Fluxgate magnetometer (Russell et al., 2016). All observations (magnetic field and spacecraft position) used in this study are averaged over 1-minute intervals and examined in the geocentric solar magnetospheric (GSM) coordinates.

We use time series of SuperMAG Auroral Electrojet (SML) index, solar wind and interplanetary magnetic field (IMF) data at 1-min time resolution from the SuperMAG database (https://supermag.jhuapl.edu/indices/, Gjerloev (2012)). The solar wind data on the SuperMAG site come from the OMNI database (https://omniweb.gsfc.nasa.gov/). The list of isolated substorm onsets comes from Ohtani and Gjerloev (2020), who identified substorm onsets using the SML index. During the period from 2016 to 2020, there were 5,077 substorms identified, and we found the substorm peak times (corresponding to the minimum SML) and end times (when SML recovers to above $-100$ nT) (see Kumar et al. (2024) for details).

In order to assess the magnetopause location as function of the solar wind parameters, we use formulation introduced by Shue et al. (1998) that gives the position and shape of the magnetopause in the form:

$$r = r_0 \left[ \frac{2}{1 + \cos\theta} \right]^{\alpha} \tag{1}$$

$$r_0 = \left[ 10.22 + 1.29 \tanh\left( 0.184(B_Z + 8.14) \right) \right] P^{-1/6.6} \tag{2}$$

$$\alpha = (0.58 - 0.007 B_Z) \left[ 1 + 0.24 \ln(P) \right], \tag{3}$$

where $r$ is the radial distance from the Earth and $\theta$ is the solar zenith angle computed from the positive $X_{GSM}$-axis. The parameter $r_0$ gives the standoff distance at the subsolar point, and $\alpha$ determines the level of tail flaring.

## 3 Magnetopause observations

All magnetic field observations in this analysis are presented in GSM coordinates. We subtracted the internal geomagnetic field by employing the International Geomagnetic Reference Field (IGRF-13) model (Alken et al., 2021), i.e., external magnetic field $\Delta B_i = B_i^{obs} - B_i^{IGRF}$, where $i = X, Y, Z$ respectively. The IGRF subtraction was done to reveal the small-scale variations in the magnetic field due to substorm processes particularly in the vicinity of the Earth, where the impact of the internal dipole





field on the overall magnetic field is significantly pronounced. Although the internal field is small in the magnetotail, we subtracted the IGRF from all observed $B_Z$ values for the sake of consistency.

We examine the external magnetic field using combined datasets from spacecraft (THEMIS-A,D, E, RBSP-A, B, and MMS-

1) during different phases of 5077 substorms and for five years 2016–2020. In Figure 1(a-f), we present color-coded maps illustrating the averaged magnetic field $\Delta B_Z$ (with IGRF field subtracted) in $2R_E \times 2R_E$ bins of $X$ and $Y$. The maps utilize 5 minutes data of $\Delta B_Z$ collected prior to onset (Pre-onset), after the substorm peak (Post-peak), and before the substorm end (Pre-end) for northward IMF ($a, c, e$) and southward IMF ($b, d, f$). In Figure 1, the average $\Delta B_Z$ is presented in the $X$-$Y$ (equatorial) plane of magnetosphere during substorm growth (Pre-onset), early recovery (Post-peak) and late recovery

(Pre-end) phases for northward IMF (IMF $\langle B_Z \rangle > 0$ nT, $a, c, e$) and southward IMF (IMF $\langle B_Z \rangle < 0$ nT, $b, d, f$) separately. The near-equatorial orbits of the spacecraft result in the most comprehensive data coverage being in the equatorial region (see Figure 2 in Kumar et al. (2024) ). In the five-year study period, the magnetic field measurements correspond to approximately 1502 substorms during northward IMF conditions and 3458 substorms during southward IMF conditions and 116 without both orientations. We illustrate the Earth at the center with a radius of 1 $R_E$. The region within 4 $R_E$ around the Earth is masked, as

we focus on the region outside that distance, and the black circle at 6.6 $R_E$ provides a reference to geostationary orbit. Contours of $\Delta B_Z = 0$ nT, highlighted in cyan, outline the location of magnetopause and the dayside compression region. In Figure 1 ($a, c, e$), the black curves are plotted on the average magnetic field $\Delta B_Z$ maps using the standoff distance $r_0$ and tail flaring angle $\alpha$ obtained from the empirical model by Shue et al. (1998). To plot these black curves, we initially utilize equations (2) and (3) for $r_0$ and $\alpha$, respectively. We then estimate their values near substorm onset, peak, and end times from Figure 3 ($j$-$o$)

for strong northward IMF, where strong northward IMF is defined as IMF $B_Z > 5$. Using values of $r_0$ and $\alpha$ around onset, peak, and end times of substorms, we estimate radial distance $r$ from equation (1) and finally calculate the positions $x_s$ and $r_s$ using $x_s = r * cos(\theta)$, $r_s = r * sin(\theta)$. It is interesting to see that $x_s$ versus $r_s$ curves (black) plotted in Figure 1 ($a, c, e$) pass through the thin boundary between the yellow and green colours (over the cyan curve) and represents the average location of the magnetopause, the outer boundary of magnetosphere. The red dashed curves in Figure 1 ($a, c, e$) are plotted exactly in the

same manner as black curves but for northward IMF, which is defined as IMF $B_Z > 0$. For these curves, we first estimate $r_0$ and $\alpha$ from figure similar to Figure 3 ($j$-$o$) (not shown) but for northward IMF. The red curves in Figure 1 ($a, c, e$) pass through the yellow color (magnetosheath, just outside the magnetopause).

Figures 1 ($b, d, f$) are plotted in the same way as Figures 1 ($a, c, e$) representing a color-coded map of averaged magnetic field $\Delta B_Z$ from growth to recovery end phases of substorm but for southward IMF (IMF $B_Z < 0$). In the Figures 1($b, d,$

$f$), the magnetopause is more clearly identified between yellow and green colours since both the colours are more darker in these panels as the number of data points (as well as number of substorms) available are more for southward IMF than those during northward IMF. The black curves plotted (same as in Figure 1 $a, c, e$) using Shue et al. (1998) model over the averaged magnetic field $\Delta B_Z$ maps for strong southward IMF (IMF $B_Z < -5$) pass through the boundary between the yellow and green colours also confirmed (as in the Figure 1 ($a, c, e$)) the location of outer boundary of magnetosphere. Similar to Figure 1 ($a, c,$

$e$), the plotting of red dashed curves (for southward IMF (IMF $B_Z < 0$)), and contour curves (for IMF $B_Z = 0$)) follows the same methodology.



Note that the in-situ measurements show an asymmetry in the flanks near the terminator. Their occurrence can be attributed to certain nonlinear effects at the bow shock and would be more prominent if the data were sorted according to the Parker spiral orientation ($B_X$ and $B_Y$) (out of the scope of this work), which would allow separation of the quasi-parallel and quasi-
perpendicular shocks in the dawn and dusk sectors.

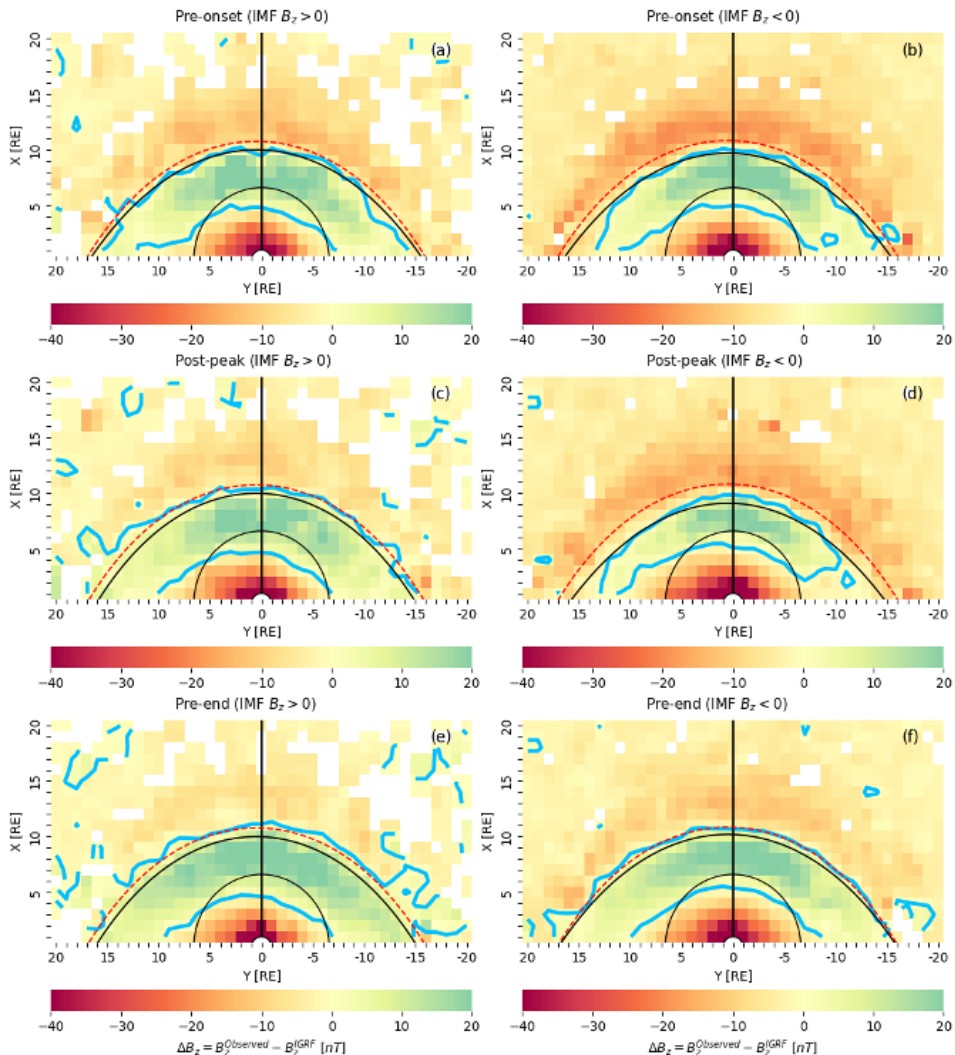

**Figure 1.** Average magnetic field $\Delta B_Z = B_Z^{obs} - B_Z^{IGRF}$ in the equatorial plane averaged over 5 min before the substorm onset (pre-onset, $a, b$), 5 min after the peak (post-peak, $c, d$), and 5 min before the end (pre-end, $e, f$) for northward IMF ($\langle B_Z \rangle$>0 nT, $a, c, e$) and southward IMF ($\langle B_Z \rangle$>0 nT, $b, d, f$) are shown separately. The black (red dashed) curves show the magnetopause location plotted using Shue et al. (1998) model with IMF $B_Z > 5$ (0) nT for northward IMF (left panels) and with IMF $B_Z < -5$ (0) nT for southward IMF (right panels). The cyan curves are plotted for the $\Delta B_Z$=0 nT.





Figure 2 shows difference maps indicating the time evolution of dayside $\Delta B_Z$ averaged for $2R_E \times 2R_E$ bins in $X$ and $Y$ during substorm onset to peak (Pre-peak – Pre-onset), around the substorm peak (Post-peak – Pre-peak), and from the substorm peak to end of the recovery phase (Pre-end – Post-peak) during northward IMF ($\langle B_Z \rangle > 0$) ($a$, $c$, $e$) and southward IMF ($\langle B_Z \rangle < 0$) ($b$, $d$, $f$). Each panel shows color-coded 2D difference map of a 5-min average data of $\Delta B_Z$ with positive values (indicating an increase in the magnetic field) displayed in red, and negative values (indicating a decrease in the magnetic field) shown in blue colors. The black (cyan dashed) curves are identical to those in Figure 1. These curves in the left panels (Figure 2 $a$, $c$, $e$) represent cases with northward IMF, specifically with IMF $B_Z > 5$ nT (0 nT), while the right panels (Figure 2 $b$, $d$, $f$) depict cases with southward IMF, corresponding to IMF $B_Z < -5$ nT (0 nT).

The difference maps for the expansion phase (Figures 2$a$, $b$) demonstrate that during this phase, the magnetic field outside the magnetopause in the magnetosheath increases (shown by red colors), more prominently in the northward IMF than in the southward IMF case. As the magnetosheath field is created by the shocked IMF, this is a reflection of an IMF maximum at the substorm onset time.

The field inside the dayside magnetosphere shows more complex behavior. For northward IMF (Figure 2 $a$), between the magnetopause and geostationary orbit, the dayside field change is predominantly negative, but inside geostationary orbit the field change is mildly positive. This would be consistent with an enhancement of the ring current in that sector, with field enhancement inside the current peak and field reduction outside of it. The bipolar structure, could be interpreted as R2 currents, is clearly discernible in that case.

We also point out that there is a bipolar structure with field increase inside geostationary orbit and field decrease outside of it in the morning sector (Figure 2 $a$), and the opposite changes in the evening sector near the terminator. This may be associated with the enhancement of the Region 2 current pattern that would have an effect opposite to that of the substorm current wedge.

For the southward IMF case (Figure 2 $b$), the dayside field is strongly negative under black (dashed cyan) curve, could be implying a strong ring current enhancement. As the field depression is negative throughout the region, the ring current peak is likely closer to the Earth, as particles under southward IMF and stronger convection have access to closer drift paths around the Earth. The bipolar structure interpreted as R2 currents is not visible for the southward IMF case.

As the substorm reaches its peak and the recovery starts, the positive field change outlines the magnetopause, indicating an inward motion of the magnetopause (as the internal field is larger than the magnetosheath field, Figures 2 $c$, $d$). Other changes inside the magnetosphere are mostly small.

The substorm recovery phase (Figures 2 $e$, $f$) causes a strong signal around the magnetopause for both northward and southward IMF cases, implying further inward motion (compression) of the magnetopause.

For both cases, the dawn and dusk fields are strongly enhancing from inside geostationary orbit out to the magnetopause.

The field continues to increase around the substorm peak time, with mostly red colors indicating further compression. Moreover, the field continues to increase strongly beyond the substorm peak, as demonstrated by strongly positive (dark red) values during the late recovery phase as shown by in-situ measurements (Figure 2 $e$, $f$). However, the Shue magnetopause exhibits a relaxation of the magnetic field (moving outward) from post-peak to pre-end, with its position changing from 10.24 to 10.27 $R_E$ (Table 1).





Figure 2 $b$ depicts that from Pre-onset to Pre-peak, the dayside magnetic field is in a relaxing state and experiences a decrease, as indicated by mostly blue colors, i.e., the magnetopause exhibits sunward motion or expansion.

This behavior is the opposite of the Shue magnetopause, which shows a compression (albeit very small) of the magnetopause from Pre-onset to Peak, with its position changing from 10.32 to 10.30 $R_E$ (Table 1). Around the substorm peak (from Pre-
peak to Post-peak), the field increases, indicated by mostly red colors, signifying slight Earthward motion of magnetopause. This pattern aligns with the empirical model results illustrated in Figures 1 $c, d$. The field continues to increase strongly from the substorm peak to the recovery end, as indicated by strongly positive (dark red) values of the magnetic field during the recovery end (Figure 2 $f$). Conversely, similar to the case of a northward IMF, the Shue magnetopause exhibits a relaxation of the magnetic field from post-peak to pre-end, with its position changing from 10.30 to 10.34 $R_E$.

The difference maps (Figures 2 $b$, $d$, $f$) show that during the expansion phase, the dayside magnetospheric field is reduced (predominantly blue colors), indicating outward motion of the magnetopause. The changes around the substorm peak time are predominantly positive, indicating further compression of the field (consistent with the empirical model results), and even more strongly positive during the recovery phase (opposite to the results from empirical model).

## 4  Superposed Epoch Analysis

Superposed epoch analysis is a statistical technique used to identify patterns in time series associated with specific events. The method allows examination of average system response centered around the zero epoch. We use three zero epoch as substorm onset (SML onset), substorm peak (SML minimum), and substorm end (SML recovery to above $-100$ nT).

Figure 3 displays the superposed epoch analysis for strongly northward IMF defined as $\langle B_Z \rangle > 5$ nT during the epoch period from substorm onset to recovery end. The panels show the observed SML index, IMF $B_Z$, solar wind dynamic pressure $P_{dyn}$,
and the magnetopause model parameters $r_0$, $\alpha$ using a 240-min time window around the zero epoch (onset, peak and end) times. The blue (red) curves represent the median (mean), the vertical black dotted lines show the zero epoch. The shading indicates the interquartile range between 25% and 75%.

In Figures 3 $a$-$c$, the SML exhibits a rapid decline or the initiation of a negative bay at substorm onset, reaching its minimum value with a peak magnitude around $-250$ nT. Subsequently, it ascends towards the pre-onset level (above $-100$ nT) by the
end of the substorm. The duration from substorm onset to peak (expansion phase) is approximately 40 minutes, and from peak to substorm end (recovery phase) is about 70 minutes. As one would expect during northward IMF conditions, this dataset comprises small, relatively short-lived substorms.

Figures 3 $d$–$f$ display that IMF $B_Z$ started to increase a few minutes before the substorm onset, indicating that the substorm onset was associated with a further enhancement of the northward IMF component. The peak of the northward IMF is coinci-
dent with the substorm and SML activity peak. The IMF magnitude starts to decrease prior to the end time and continues to do so after the the recovery phase ends.

The solar wind dynamic pressure $P_{dyn}$ results (Figures 3 $g$-$i$) reveal only very weak changes near the substorm onset time. In the minutes leading to the substorm onset, there is a discernible decrease in the average magnitude of $P_{dyn}$ that reaches its



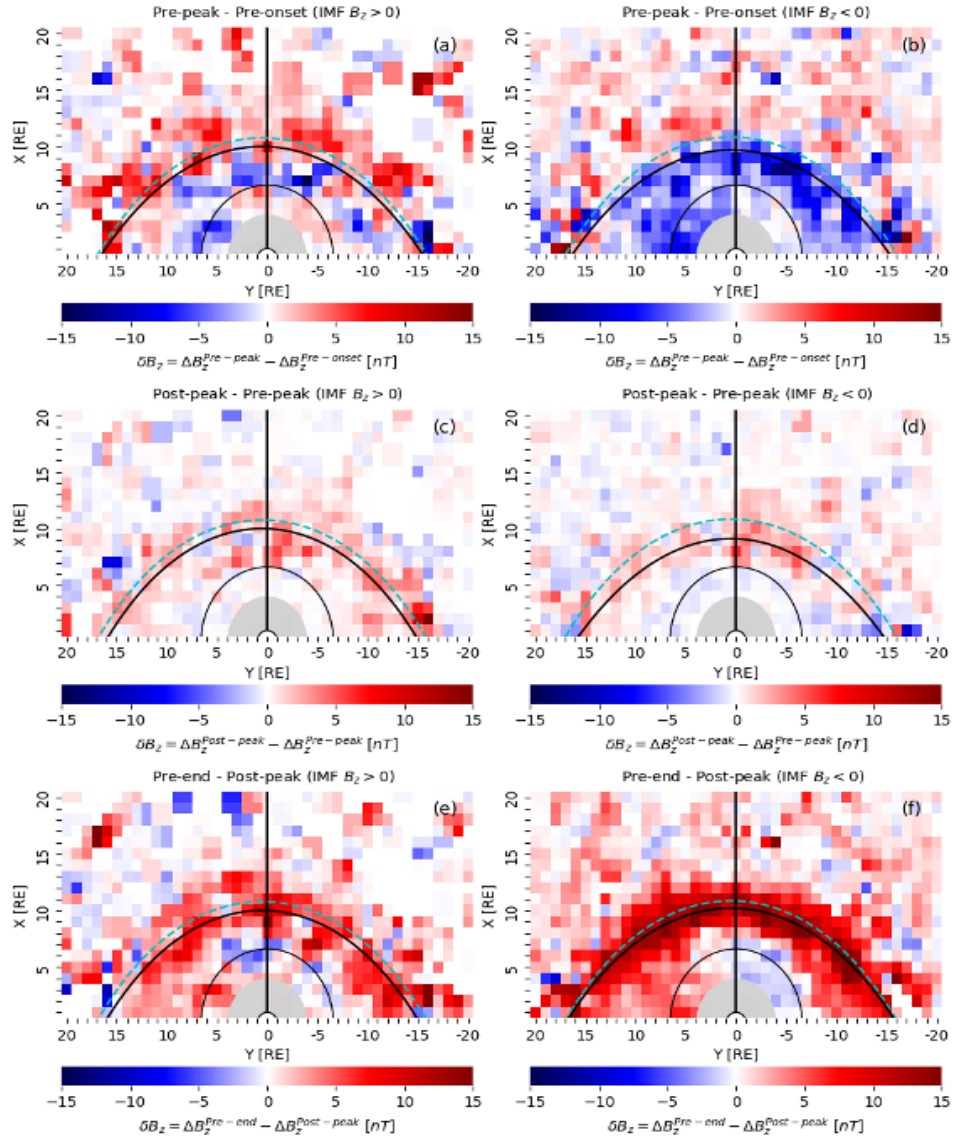

**Figure 2.** Average differences $\delta B_Z = \Delta B_Z^{pre-peak} - \Delta B_Z^{pre-onset}$, $\delta B_Z = \Delta B_Z^{post-peak} - \Delta B_Z^{pre-peak}$, and $\delta B_Z = \Delta B_Z^{pre-end} - \Delta B_Z^{post-peak}$ indicating changes in magnetic field from substorm onset to peak, around the peak, and from peak to recovery end during northward IMF ($\langle B_Z \rangle > 0$ nT, $a$, $c$, $e$), and during southward IMF ($\langle B_Z \rangle < 0$ nT, $b$, $d$, $f$). The black (cyan, dashed) curves are the same as in the Figure 1 and show the magnetopause location plotted using Shue et al. (1998) model with IMF $B_Z > 5$ (0) nT for northward IMF (left panels) and with IMF $B_Z < -5$ (0) nT for southward IMF (right panels).

lowest point at the onset. During the substorm peak, the average magnitude of $P_{dyn}$ remains nearly constant and persists at the
same level even beyond the recovery phase's end. Figures 3 $j$-$l$ show the magnetopause subsolar location ($r_0$) evaluated using





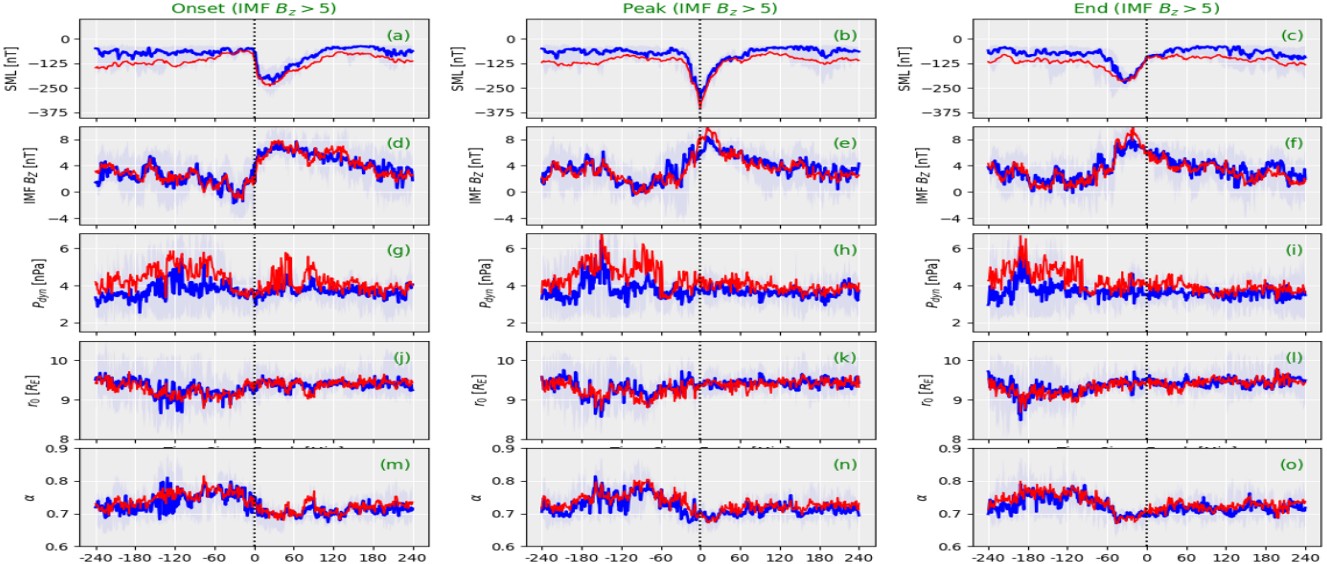

**Figure 3.** Superposed epoch analysis (median (blue), mean (red), interquartile range (shaded)) of the (*a-c*) SML index, (*d-f*) IMF $B_Z$, (*g-i*) dynamic pressure $P_{dyn}$, and magnetopause location parameters (*j-l*) $r_0$, (*m-o*) $\alpha$ for strongly northward IMF ($\langle B_Z \rangle > 5$ nT). Three zero epoch times are used: (left) substorm onset, (center) substorm peak, and (right) substorm end.

Equation 2 which gives the standoff distance at subsolar point as function of the upstream solar wind dynamic pressure $P_{dyn}$ and the IMF $B_Z$. Overall, the changes in the subsolar point location are small during northward IMF. However, the subsolar distance increases toward the end of the growth phase and has a small peak at the substorm onset time. This demonstrates the significant reliance of $r_0$ on solar wind dynamic pressure, as it exhibits a slight increase during a slight decrease in solar wind pressure, despite an increase in IMF $B_Z$ near onset. Even after the onset, $r_0$ follows the trends in solar wind pressure, continuing beyond the substorm end despite variations in IMF $B_Z$ near the peak and recovery end. Figures 3*m-o* display the results for the tail flaring parameter ($\alpha$, Equation 3). The flaring exponent starts to decrease before the substorm onset, indicating that there is a reduction in the tail flaring angle at the same time as the subsolar point is moving away from the Earth. The flaring exponent value is at minimum at the substorm peak, after which it starts to increase slightly again.

Figure 4 displays the results of superposed epoch analysis of SML, IMF $B_Z$, $P_{dyn}$, and the magnetopause location parameters $r_0$, $\alpha$ similar to Figure 3 but for strongly southward IMF ($\langle B_Z \rangle < -5$ nT during the interval from substorm onset to the recovery end. The top row of Figure 4 (*a-c*) shows clear growth, expansion and recovery phase signatures in the SML index. The duration of expansion phase is nearly 120 minutes (Figure 4a) for substorms during southward IMF, which is much longer than the expansion phase for substorms during northward IMF. The substorms are very strong (higher amplitude, $\sim -750$ nT) and their recovery time scale is significantly longer ($\sim 140$ min, Figure 4a,b) compared to substorms during northward IMF.

Figures 4 *d-f* show a quite different pattern from the northward IMF case: During the growth phase, IMF $B_Z$ decreases to reach a minimum at substorm onset, without a signature of northward turning at that time. The IMF $B_Z$ increase starts prior





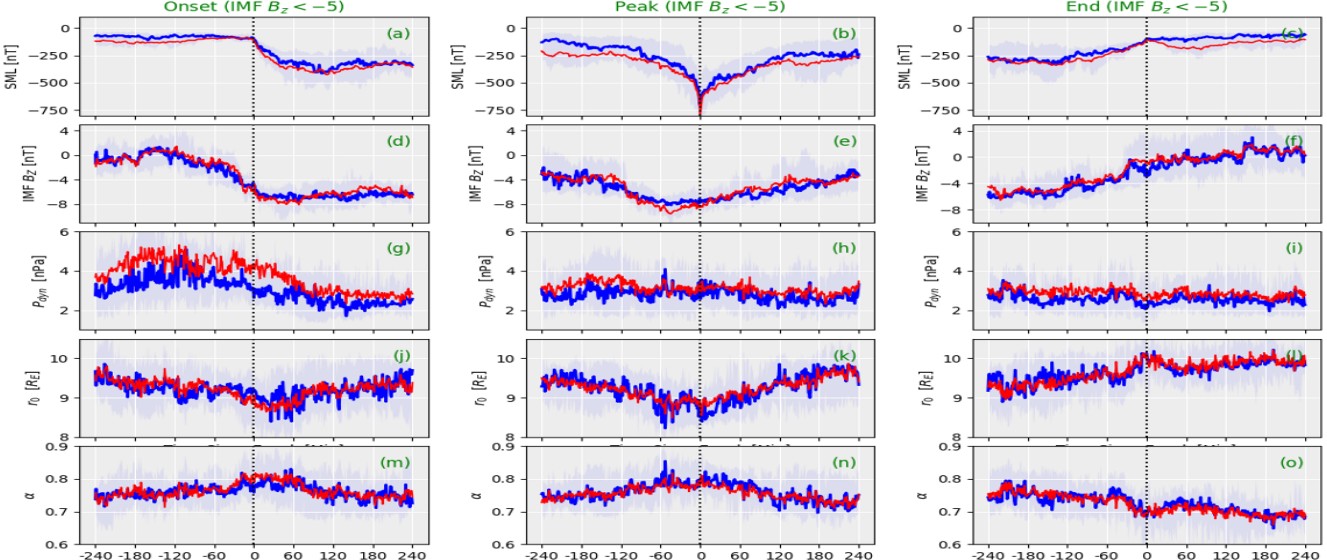

**Figure 4.** Superposed epoch analysis with the median (depicted in blue), mean (shown in red), and interquartile range (highlighted) of various parameters: the SML index (*a-c*), IMF $B_Z$ (*d-f*), solar wind dynamic pressure $P_{dyn}$ (*g-i*), magnetopause location parameters $r_0$ (*j-l*) and $\alpha$ (*m-o*), specifically focusing on instances of strongly southward IMF ($\langle B_Z \rangle < -5$ nT). The analysis is conducted at three distinct zero epoch times: substorm onset (left), substorm peak (center), and substorm end (right).

to the peak time, without a clear timing in relation to the substorm phases, and continues throughout the end of the substorm. The IMF changes are smooth and broad, indicating that they are not directly associated with the substorm timing.

The dynamic pressure trends (Figures 4 *g-i*) show a pressure decrease during the growth phase, while there are no clear trigger signatures either at onset, peak, or end times (note a slight pressure minimum at the substorm end).

Figures 4 *j-l* show the decrease of the standoff distance during the substorm growth phase due to the flux erosion from the dayside driven by dayside reconnection. The subsolar distance has a minimum at the peak of the substorm, and starts a gradual increase that continues throughout the recovery phase. The end time is associated with a localized peak in the standoff distance.

Figures 4 *m-o* show the results for the tail flaring ($\alpha$). The flaring parameter increases during the growth phase and has a broad peak during the expansion phase (between onset and peak time). The flaring parameter has a minimum at the end of the substorm, coincident with the peak in the standoff distance.

## 5   Empirical model

The Shue et al. (1998) model is an empirical model developed through a statistical analysis of an extensive dataset of mag-
netopause crossings, considering the pressure exerted by the incoming solar wind on the magnetosphere and the southward component of the IMF, which plays a pivotal role in the dayside reconnection process at the magnetopause. This model pre-



dicts the magnetopause's location as a function of two input parameters ($P_{dyn}$ and IMF $B_Z$). Based on the predicted location, the model offers an estimation of the magnetopause shape. Due to its simplicity and accuracy under specific solar wind conditions, this model has become a widely utilized tool in space weather research and magnetospheric simulations and therefore,

it is employed in this study to estimate the average location of the magnetopause and shape at substorm onset, peak, and end times. The Shue model is solely parameterized by solar wind parameters and was not originally intended to account for substorm variations. However, despite this limitation, its predictions serve as valuable contextual information for interpreting the statistics derived from the magnetopause.

In the panels of Figure 1 and 2, each figure exhibits black and red-cyan dashed curves are plotted over the average 2D maps

of $\Delta B_Z$. These curves are generated using the standoff distance $r_0$ and flaring angle $\alpha$ parameters from the Shue et al. (1998) model at the times of substorm onset, peak, and end. The values of these parameters at various substorm timings are derived from superposed epoch analysis (see Figures 3 and 4). Utilizing the values of $r_0$ and $\alpha$ around substorm onset, peak, and end times, we calculate the radial distance $r$ using equation (1) and then determine the positions $x_s$ and $r_s$ through $x_s = r\cos(\theta)$ and $r_s = r\sin(\theta)$. When these $x_s$ versus $r_s$ curves are plotted on the average 2D maps of $\Delta B_Z$ for strongly northward or

strongly southward IMF ($|B_Z| > 5$), they appear in black in Figure 1 and Figure 2. Additionally, we show northward ($B_Z > 0$) or southward IMF ($B_Z < 0$) model results by red and cyan dashed curves.

In these figures, the Shue magnetopause aligns closely with the zero contour (cyan curves in Figure 1), particularly during onset and peak, signifying the average magnetopause location for strongly northward/southward IMF conditions. During substorm end times, the cyan curve representing the magnetopause boundary indicates outward movement is in line with the Shue

model. This indicates that during substorms end, the magnetopause is slightly further away from the Earth than predicted by the Shue et al. (1998) model. However, the model curve (red dashed) consistently failed to predict the magnetopause boundary for both northward and southward IMF ($|B_Z| > 0$) conditions at all substorm timings and it traverse far from the $\Delta B_Z = 0$ boundary. The differences between red dashed and black curves are small, but more prominent just before substorm onset (Pre-onset) and after the peak (post-peak) of the substorm during southward IMF Figure 1 $b, d$. During the southward IMF at

recovery end of the substorms, the Shue et al. (1998) model (res dashed curve) is observed to align closely with the cyan curve, while the black curve passing slightly inside of it. The difference between the black and red curves is very small in comparison to the curves near substorm onset and peak.

The subsolar distances in the Shue et al. (1998) model for various superposed epoch results are presented in four rows in the table, showing their values for positive, strongly positive, negative, and strongly negative average IMF $B_Z$, respectively. Each

row indicates times just before substorm onset, after substorm peak, and before substorm end.

In each case, the magnetopause is shown to be closest to the Earth at the peak of the substorm, recovering outward during the recovery phase – in line with the in situ measurements (Figure 1 $e, f$). Furthermore, comparing the Shue magnetopause location during strongly northward and southward IMF, it is evident that the compression of magnetopause is most pronounced for a strong southward IMF and at the substorm peak.



| Substorm phase | Subsolar distance $r_0 [R_E]$ | | | |
| --- | --- | --- | --- | --- |
| | $B_Z > 0$ | $B_Z > 5$ nT | $B_Z < 0$ | $B_Z < -5$ nT |
| Onset | 10.25 | 9.49 | 10.32 | 9.19 |
| Peak | 10.24 | 9.47 | 10.30 | 8.59 |
| End | 10.27 | 9.49 | 10.34 | 9.69 |

**Table 1.** Subsolar distances in the Shue et al. (1998) model for the different superposed epoch results. The columns show the values for positive, strongly positive, negative, and strongly negative average IMF $B_Z$, respectively. The rows indicate times just before substorm onset, after subtorm peak, and just before substorm end.

## 6   Discussion and Conclusions

In this study, we explore variations in the average position of the magnetopause during different phases of magnetospheric substorms. The average location of the magnetopause is determined through magnetic field observations collected by space missions such as THEMIS-A, D, E, RBSP-A, B, and MMS-1 over a five-year period from 2016 to 2020. For the estimation of magnetopose location, we employ the empirical model for magnetospheric shape and size proposed by Shue et al. (1998), incorporating OMNI solar wind data, specifically solar wind dynamic pressure ($P_{dyn}$) and IMF $B_Z$, throughout the study period. A list of substorm onsets, identified by a change in the SML index, were obtained from the work of Ohtani and Gjerloev (2020). In order to investigate changes in the magnetopause location during different substorm phases, we identified the peak and end times of each substorm in a subset of 5,077 substorms identified within this study period. The initial step involves computing the average of the IMF $B_Z$ for each substorm (from onset to recovery end). Subsequently, we filter substorms based on their occurrence during northward IMF ($B_Z > 0$) and southward IMF ($B_Z < 0$). By combining magnetic field measurements from all satellites over the five-year duration, we generate average 2D maps of the observed $\Delta B_Z$ for northward-southward IMF during distinct substorm phases, including pre-onset, post-peak, and pre-end (Figure 1).

The 2D average maps (Figure 1) clearly outline the dayside magnetopause as a very thing boundary between positive (green) and negative (yellow) $\Delta B_Z$, as the magnetosheath field is weaker than the dipole, and the dayside magnetospheric field is stronger than the dipole due to the compression by the solar wind dynamic pressure. The magnetopause appears to be compressed near substorm onset and peak for both northward and southward IMF conditions (Figure 1 $a, c, b, d$), and exhibits an outward movement just before the recovery end (Figure 1 $e, f$). However, Figure 1 $f$ demonstrates an alignment of the cyan and red dashed curves for southward IMF, indicating a small outward motion of the magnetopause but less in contrast to the corresponding figure for northward IMF.

The increase in dayside magnetic field (magnetopause compression) from the expansion to the recovery phase during both northward and southward IMF orientations can be seen clearly in the difference maps of $\Delta B_Z$ (Figure 2). Examination of the field changes during the expansion and recovery phases could be revealed by the changes in the R1/R2 current patterns. The magnetospheric field differences clearly show the magnetopause motion.





The magnetopause motion during dayside reconnection is associated with generation of field-aligned Region 1 and 2 currents (Birkeland, 1908; Iijima and Potemra, 1976). Coxon et al. (2014) explored the role of the solar wind, IMF, dayside and nightside reconnection in generating field-aligned currents within the interconnected magnetosphere-ionosphere system. They show that these currents respond to the combined effects of dayside and nightside reconnection, and that R1 currents tended to be higher than R2 currents during dayside reconnection.

Additionally, we performed superposed epoch analysis of solar wind and magnetospheric parameters to examine their behavior for strong northward and southward IMF at various substorm timings, including onset, peak, and end phases (Figure 3 and Figure 4). We used the results to compute the average magnetopause location through standoff distance $r_0$ and shape by tail flaring angle $\alpha$ from magnetopause model by Shue et al. (1998) which is derived on the basis of solar wind dynamic pressure and IMF $B_Z$.

Our superposed epoch analysis of the solar wind and IMF parameters reveal a clear correlation between substorm onsets and changes in the IMF direction (Figure $3d - f$ and Figure 4 $d - f$); this was true for both substorms occurring during strong northward and southward IMF. Which show that our findings are consistent with earlier research, which has shown that substorm onsets are associated with intervals of southward IMF (Kokubun, 1972; Wild et al., 2009), as well as with the northward turning of the IMF (Mcpherron et al., 1986; Sergeev et al., 1986; Hsu, 2003).

The values of subsolar point $r_0$ is estimated near substorm onset, peak and end from Figure 3 ($j - l$) and Figure 4 ($j - l$) and are shown in the Table (see Table 1) for more clarity. From the table it is clear that the subsolar point moves toward the Earth from substorm onset to the peak and is closest to the Earth at the peak of the substorm for both IMF $B_Z > 0$, 5 and $B_Z < 0$, -5. It then moves outward from the peak to the substorm recovery end, such that it is farther from the Earth at the end of the substorm than it was at substorm onset form all IMF $B_Z$. The Shue et al. (1998) predicts the behavior of magnetopause similar to shown in average maps of $\Delta B_Z$ (Figure 1) near substorm onset, peak and end for IMF $|B_Z| > 0$.

The empirical model results for northward-southward IMF condition show that the model magnetopause for northward IMF (red dashed curves) is slightly further out than seen in the in situ data (cyan curves). Whereas, the model values for strongly northward-southward IMF (black curves) produces the best fit with the in situ measurements and give more accurate location of magnetopause from pre-onset to pre-end and same can be seen the in-situ data which indicate compression during onset and peak and relaxation at the recovery end, as illustrated by the cyan curves in Figure 1.

In summary, we utilize an extensive dataset from multi-satellite observations and Shue et al. (1998) model to demonstrate the changes in magnetopause position under the influence of northward-southward IMF and internal magnetospheric process like substorms and we observed that:

(1) The analysis of 2D equatorial maps of the average $\Delta B_Z$ illustrates the changes in dayside magnetic field during substorm timings and the average position of magnetopause by a thin boundary between green and yellow colors (cyan curves) under both northward and southward IMF conditions.

(2) From superposed epoch analysis of IMF $B_Z$, the signatures of substorm occurrence are found during the period of northward as well as southward IMF.



(3) The decrease in the values of $r_0$ suggest an earthward movement of the magnetopause and its closest approach to Earth near the substorm peak time during strong southward IMF ($B_Z < -5$).

(4) Shue et al. (1998) model accurately predicts the average magnetopause location during substorm timings, particularly for strong northward and southward IMF orientations (IMF $|B_Z| > 5$).

(5) The differences between the substorm-time values and the average conditions indicate that the internal magnetospheric state impacts the location of (and likely processes at) the magnetopause. This may implicate a more complicated relationship between geomagnetic activity and the solar wind driver than illustrated by solar wind - based coupling functions (Newell et al., 2007).

## 7   Open Research

All data used in this study are available through the NASA Space Physics Data Facility (SPDF, or cdaweb.gsfc.nasa.gov/pub/data, and the SuperMAG website (https://supermag.jhuapl.edu/indices/).

*Acknowledgements.*  We acknowledge the substorm timing list identified by the Newell and Gjerloev technique Ohtani and Gjerloev (2020), the SMU and SML indices Newell and Gjerloev (2011); and the SuperMAG collaboration Gjerloev (2012). This material is based upon work supported by the National Aeronautics and Space Administration under Grant/Contract/Agreement No. 80NSSC21K1675 issued through the Heliophysics Supporting Research Program.

*Author contribution.* All authors contributed equally to this paper.

*Data availability.* All data used in this study are available through the NASA Space Physics Data Facility (SPDF, or cdaweb.gsfc.nasa.gov/pub/data, and the SuperMAG website (https://supermag.jhuapl.edu/indices/).

*Competing interests.* The authors declare that they have no conflict of interest.





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
