# Peer review of "Statistical Analysis of Magnetopause Response During Substorm Phases"

_EGUsphere, 2024_

## Author Response (AR1)

Response to Reviewer 1:

We would like to thank the reviewer very much for providing insightful comments which guided us strengthen our manuscript. Based on the comments of the reviewer, we have revised the manuscript. Please find below our detailed responses to each of reviewer's comment.

One could also argue that your title is too generic or even misleading. After all, your own observations exclusively focus on the magnetic field, but (somewhat surprisingly) ignore plasma effects like e.g., the role of the solar wind pressure, possible magnetopause boundary layers, transport mechanisms etc. This should be reflected in the title.

.............

We have now updated the title to better align with the focus of the current study.

............

Major concerns
————————

My major concern about the paper is the methodology (or possibly the explanation or my understanding of the methodology). Rather than actually observing magnetopause crossings, you seem to rely on deviations between the measured field and a model field (IGRF). Plasma or particle aspects, e.g, plasma or thermal pressure are largely ignored. This raises several questions that should be addressed:

1) From my understanding, you identify the magnetopause as the region where deltaBz is 0, i.e., the measured and modeled magnetic fields are identical along the Z axis (line 106, and marked with cyan contours in the plot). How does this signify the magnetopause? Does it not only show where the model (IGRF) agrees with measurements? Some elaboration is needed here.

.........

We acknowledge the referee's observation that at the location where $\Delta B_Z = 0$, the observed and modeled magnetic fields are identical, i.e., $B_Z^{Observed} = B_Z^{IGRF}$ at the magnetopause.

To avoid confusion, we have now included plots of the observed $B_Z$ without the IGRF subtraction in the revised manuscript. This addition will provide a clear representation of the magnetic field variations and the location of the magnetopause.

Our primary objective was to determine the average location of the magnetopause during substorm phases, and whether or not we subtracted the IGRF did not affect the behavior of the magnetopause during these events.

..........

2) Your figure 1 shows maps of these magnetic field deviations. Here, I am critical to the use of IGRF outside of the magnetopause. Your map covers up to 20 Re sunward, i.e., well outside the typical magnetopause and well into the magnetosheath, or possibly into the pristine solar wind plasma region. How valid is the IGRF here ?

...........

Beyond $X > \sim 10 \ R_E$ on the dayside, the internal magnetic field is zero, and only the external (modeled) magnetic field is displayed. As our analysis focuses on the location of magnetopause, we are not concerned with the external field in the solar wind, and we did not discuss it in this study.

Although, earlier we subtracted the IGRF from all observed BZ values for the sake of consistency only. But now in the revised manuscript we plotted observed Bz without IGRF subtraction in Figure 1, 2 and obtained same results.

...........

3) The color scale in Figure 1 is not really ideal to illustrate deviations, but in my eyes, the most pronounced feature is the red region (i.e., largest negative deviation between measured field and IGRF) close to Earth. This is unclear to me. I would expect that the best agreement would be close to Earth where IGRF is a much better representation of the model field than e.g., in the solar wind. I know that this is outside your focus region, and you mask it out in later plots. Still, this is an eye catcher of this plot and needs to be explained (even if you do not show it visually).

.............

Sorry for the confusion. The region near the Earth appears to be predominantly influenced by the modeled field (IGRF), as indicated by the presence of negative magnetic field values. The exact reason for this dominance is currently unclear and requires further investigation.

In the revised manuscript, we have chosen to mask the region within 4 $R_E$ (Earth radii) around the Earth, as depicted in Figures 1 and 2 of revised manuscript. This decision was made because this particular region contains high dipole field values,

which are not the primary focus of our study. By masking this region, we can direct our attention and analysis towards the specific aspects that are relevant to our research objectives.

To correctly include the differences at distances close to the Earth where the field is large would require using relative errors (e.g., difference divided by the model value) which would not overweight the strong-field region.

..............

4) There are also a couple of inconsistencies regarding the position of the magnetopause and the underlying reason for this. I think the effects of motion due to reconnection (i., e the actual reconnection process) and magnetopause motion due to pressure imbalance are mixed up at times, or not fully consistent with existing knowledge. See details below.

..............

There are a number of studies who identified inward motion of magnetopause during southward/northward IMF. The motion of the magnetopause is a dynamic interplay between the IMF, particularly its Bz component, and the solar wind dynamic pressure. Changes in these factors lead to the contraction or expansion of the magnetopause. $tsyganenko and Sibeck(1994), fairfield(1971), s$

............

Minor issues
————

During reading of the paper I came across a number of minor issues that should be addressed. Number given refer to line numbers the draft.

9: '..the magnetopause undergoes a significant compression..'. I doubt that the dayside magnetpause can be much compressed, i.e., get thinner; I think you mean motion or inward/outward displacement here.

............

Sorry for the confusion. we were trying to say that the magnetopause move significantly towards Earth. Now we rephrased the sentence in the revised manuscript.

..............

20: You should specify 'dayside' magnetopause here; it is unlikely that the flanks of the magnetopause can be shifted to geosynchronous distance. Also, I do not think that erosion due to reconnection can significantly move the magnetopause. The inward motion is more likely cause by enhanced dayside pressure; In the conceptual Dungey cycle, flux just circulate, so flux tubes eroded in the dayside are replenished by the return convection.

..........

Sorry for the confusion. We have now corrected the sentence and provided the appropriate references.

..........

49: I suggest to remove 'boundary'. I think you mean magnetopause (which is a boundary in itself, and should not be confused with magnetopause boundary layers, which are layers of plasma adjacent to the magnetopause. You do not use plasma observations in your study) here.

...........

Now we have removed the misleading word in revised manuscript.

...........

50: I would have emphasized that all your measurements are from the low latitudes (equatorial orbits).

...........

Yes, the orbits of the RBSP, THEMIS and MMS satellites are near the equatorial plane, which corresponds to low latitudes. These missions are designed to investigate key processes in the magnetosphere, many of which occur in the near-equatorial plane.

..........

55: The number of substorms is repeated many times (e.g.,lines 55,78,95...), but when discussing phases (lines 100-105), the numbers do not add up. Please check.

..........

We have corrected in the revised manuscript. ..........

65: What does 'higher-altitude orbit' mean?

..........

For a magnetospheric satellite in an elliptical orbit, altitude can be related to the apogee (farthest point from Earth) of the orbit. For MMS, we used near-equatorial, higher-altitude orbit with apogee 30 RE, THEMIS has apogees of 12 RE and RBSP 6 RE. Now we mentioned their apogees in the text to avoid confusion.

.........

88: repeated information. See line 73.

..........

Now we have removed the repeated sentence.

.........

91: '..small scale deviations due to substorm processes..'. Deviations from the IGRF model can also be caused buy other processes than substorms. As noted above, I also question the validity of the IGRF model close to the magnetopause.

...........

We have now plotted the observed magnetic field (without IGRF subtraction) in Figures 1 and 2 and updated the text accordingly in the revised manuscript.

..........

95: Unnecessary repetition of number of substorms.

..........

Now corrected

..........

Fig 1: I think this figure is misleading. The color coding indicate strong deviations (red color) from IGRF close to Earth (where one would expect the best agreement). Although this is not your main focus region, it may indicate a flaw in calculations or measurements, and should be investigated and explained.

...........

Now we have plotted observed Bz (with out IGRF subtraction) in Figure 1, 2 and mask the region within 4 $R_E$ in order to avoid confusion related to the strong deviations (red color) from IGRF close to Earth.

...........

139: '..during this phase..'. Which phase of the substorm is discussed here ?

..........

Now corrected.

..........

141: 'As the magnetosheath...'. I do not understand this sentenced. Particles can be reflected at shocks and boundaries, but I have never heard about reflection of a magnetic field. Likewise, I have a hard time understanding the next few lines. The discussion about the ring current, R2 currents (which, to my knowledge usually designate ionospheric field aligned currents closing at the magnetopause) is somewhat confusing. Can it be improved ?

..........

Sorry for the confusion. By the sentence "reflection of a magnetic field." we mean to say that "this is an indication of an IMF maximum....". Now we have rephrased that sentence.

We have now removed the misleading sentence related to the R2 current in the revised manuscript.

.........

158: I do not fully understand why the substorm recovery phase would imply a further compression of the magnetopause, and it is not consistent with your statement in line 8 (abstract, where you use the therm 'relaxation')

..........

Sorry for the confusion and now we have corrected that sentence in the revised manuscript. Notably, our findings show outward motion of magnetopause during substorm recover phase.

.........

169, 174: You here discuss displacements of the order of a a few 100 km. I would be very careful when interpreting such small numbers - they are most likely well below the uncertainty in methodology or model output.

...........

We agree with the reviewer's observation that the displacements of the order of a few 100 km is very small, but the tendency of magnetopause moving outward during substorm recovery phase, clearly visible in in-situ measurements (Figure **??** $e$, $f$), is supported by the Shue model as well (Table 1).

...........

203: Regarding standoff distance and Fig 3. From my interpretation, the motion is almost insignificant, and this statement is also inconsistent with the above comment and line 159 "..recovery phase..implying further inward motion..."

........

We have corrected the statement on line 159, ensuring it aligns with the statements on lines 203.

........

222: "..decrease in standoff distance due to..flux erosion..driven by reconnection". I think a better explanation is needed. The subsolar magnetopause is typically less than 1000 km thick (e.g., MMS results from Paschmann et al, JGR, 2018). The inward dispacements reported here are much larger than this, and I think it is more correct to say that a change in pressure balance, rather than the reconnection process itself cause this displacement of the magnetopause.

.......

We have removed the sentence to prevent any confusion and elaborated it more in the revised manuscript.

.........

Table 1: This table gives the impression that the standoff distance is solely governed by IMF Bz, but what about the solar wind dynamic pressure and other parameters?

.......

It is well-known that changes in the magnetopause location arise due to variations in the IMF $B_Z$, dynamic pressure, and other factors. However, its position is heavily influenced by solar wind pressure. We studied separately the variation of $r_0$ with respect to changes in solar wind dynamic pressure. For pressures $\leq 2$ nPa, $r_0$ is approximately 10.7 $R_E$ during pre-onset and 10.73 $R_E$ near the substorm end. For higher pressures ($\geq 5$ nPa), $r_0$ is about 8.6 $R_E$ during pre-onset and 8.7 $R_E$ near the substorm end. This indicates that solar wind pressure has a more significant effect on the magnetopause location than the IMF $B_Z$. However, similar to the results for IMF $B_Z$ changes, the variation in $r_0$ during substorm phases is minimal and thus figure not shown in this study.

........

278: This is the summary, but this paragraph starts with a fairly detailed interpretation of Figure 1 again , including references to single panels and details that partly repeats the earlier descriptions starting around line 95. I suggest to simplify and to synthesize the text here.

..........

Thank you for your suggestion. I have now revised the discussion and summary to make it more concise and clear. Please see the revised version in the revised manuscript.

..........

**References**

Response to Reviewer 2:

We would like to thank the reviewer very much for providing insightful comments which guided us strengthen our manuscript. Based on the comments of the reviewer, we have revised the manuscript and changes are in blue color. Please find below our detailed responses to each of reviewer's comment.

5 Major comments:

1. Overall, the method for identifying the magnetopause location is very simplistic, potentially overly simplistic. The average magnetopause location is identified using the Shue model, which is parameterized by quantities from the solar wind only and does not account for, e.g., variations in the internal magnetospheric pressure.
..........

10 Reply: We agree with the reviewer that the Shue model relies exclusively on solar wind parameters and was not initially designed to account for variations in internal magnetospheric pressure or substorms. Despite this limitation, we opted to use the Shue model in this study due to its simplicity and reliable measurement technique, enabling us to simply estimate the average distance of the magnetopause from the centre of the Earth and its shape at various substorm times. Its predictions offer valuable contextual insights for interpreting the statistics derived from the magnetopause using multi-spacecraft data.

15 Selecting to use a model based on solar wind parameters only as a reference allows us to examine the changes in the internal state of the magnetosphere during substorms, and its impacts on the magnetopause position. Without such a reference value, it would be difficult to know whether the position is affected by solar wind conditions or by substorm-associated changes in the magnetosphere. ...........

20 2. An observed magnetopause location is extracted from contours of the difference between the average measured magnetic field and the IGRF. It is not clear why this method is chosen as sharp gradients of plasma density, plasma beta, changes in the composition of minor ions, etc. are more commonly used – and arguably superior – identifiers. (While the ion composition is not measured by THEMIS, it is by MMS and Van Allen Probes). The study does not provide an adequate justification of the choice of methodology, i.e., a justification that the choice of data and models are useful for studying fundamental physical

25 processes (erosion / compression / etc?)
.........

Reply: We agree with the reviewer that methods such as magnetopause crossings, sharp gradients in plasma density and plasma beta, and changes in minor ion composition are commonly used to determine the magnetopause location. However,

30 in this study, we relied on identifying sharp gradients in the observed magnetic field, which effectively serves the same purpose. Magnetic field data, readily available from all the space missions (THEMIS, MMS, RBSP), provide a consistent and reliable means of analysis. In contrast, studying magnetopause location using ion composition data posed challenges due to its unavailability across all these spacecraft.

Additionally, to avoid confusion, we have now included plots of the observed $B_Z$ without the IGRF subtraction in the

35 revised manuscript. This addition will provide a clear representation of the magnetic field variations and the location of the magnetopause.

Using time series during magnetopause crossings to examine the magnetopause position is possible, but that methodology does not allow for binning the data with respect to substorm phase, as the crossings occur at random time intervals. The statistical method employing all data and seeing where the field changes from that dominated by magnetospheric currents to

40 one that is dominated by the IMF is the only way to discern the relatively short time intervals separating the substorm phases.
.........

3. Line 103 throughout: "In the five-year study period, the magnetic field measurements correspond to approximately 1502 substorms during northward IMF conditions and 3458 substorms during southward IMF conditions and 116 without both

45 orientations." More information is needed in this section to explain how the northward vs southward IMF categorization is performed. It is not until section 6 that I found the explanation "The initial step involves computing the average of the IMF BZ for each substorm (from onset to recovery end)".

Furthermore, the authors should justify this choice. For instance, if the IMF is southward or small during the growth phase but northward on average, erosion of the low-latitude magnetospheric field may still contribute to the magnetopause location.

50    ...........

Reply: The first step involves aggregating magnetic field measurements from all satellites over the five-year period, resulting in nearly 15 million data points when averaged over 1-minute intervals. This magnetic field data is combined with solar wind data, specifically the IMF $B_Z$ and dynamic pressure, obtained from the OMNI database and also averaged over 1 minute. We utilize a list of substorms and develop an algorithm to identify the time intervals from the onset to the end of each substorm.

55    This approach enables us to compute the average of IMF $B_Z$ for each substorm period (from onset to recovery end), offering insight into the typical value of IMF $B_Z$ during each event. We then filter the data based on IMF $B_Z$ values, distinguishing between IMF $B_Z > 0$ (northward IMF) and IMF $B_Z < 0$ (southward IMF). This allows us to estimate the number of substorms occurring under both northward and southward IMF conditions. During the study period from January 1, 2016, to December 31, 2020, we observed a total of 5,077 isolated substorms. Of these, the majority (3,458) occurred during periods of southward

60    IMF, compared to 1,502 substorms during northward IMF. Additionally, 117 substorms occurred independently of any IMF changes.

The justification for choosing substorms during northward/southward IMF $B_Z$ is that there is a strong correlation between IMF $B_Z$ and the occurrence of magnetospheric substorms. The more prolonged and intense southward IMF $B_Z$, the more energy is transferred into the magnetosphere, leading to more frequent and intense substorms. When the IMF $B_Z$ is northward,

65    the probability of substorm occurrence is lower. However, substorms can still occur, often due to other processes or prior build-up of energy in the magnetotail.

The erosion of the magnetospheric field is linked to the IMF's orientation, which could indeed impact the magnetopause location (although small) through magnetic reconnection (lines 20, 30). By separating data based on IMF $B_Z < 0$, we can monitor the magnetopause location specifically during southward IMF conditions. Similarly, analyzing data with IMF $Bz > 0$

70    allows us to examine the magnetopause location during only northward IMF, avoiding confusion between the two scenarios.

We have included this text in the revised manuscript.
    ...........

4. Fig. 1: It would be useful to understand how many magnetic field measurements / data points are available per bin. The

75    differences in the spatial distributions of the "B" parameter are very subtle from one substorm phase to the next. Is the coverage sufficient to call these differences physical?
    ..........

Reply: Each panel in Figures 1 and 2 consists of 861 bins, but the number of data points per bin varies. In the panels showing in Figures 1a, 1c, 1e for northward IMF, the number of data points in the bins ranges from 0 (lighter bins) to a maximum of

80    351 (darker bins). In contrast, the panels depicting in Figures 1b, 1d, 1f for southward IMF contain more data points per bin, with counts ranging from 0 to a maximum of 700, reflecting the higher number of substorms during southward IMF periods.

As illustrated in Figure 1, the magnetopause displacements during substorm phases are minimal. However, the outward movement of the magnetopause during the substorm recovery phase is clearly evident in the in-situ measurements (Figure 1e, 1f). This outward shift is also prominently displayed in Figure 2e, 2f, where the red color indicates an increase in the magnetic

85    field. The Shue model further supports and confirms this tendency of the magnetopause movement, as shown in Table 1. The subtle variations in the spatial distributions of the $\Delta B$ parameter from one substorm phase to another are indeed a critical aspect of our findings. We emphasize that while these differences may appear minor, they are derived from a comprehensive dataset collected over a five-year period, which includes a significant number of substorm events. This extensive dataset enhances the reliability of our observations and supports the assertion that these differences are physical rather than artifacts of limited data

90    coverage.
    ..........

Minor comments:

1. Line 36: "Substorms are transient phenomena that occur in the Earth's magnetotail, storing and releasing solar wind

95    energy through an explosive process". The second sentence in this paragraph is a more accurate introduction of substorms. It is not correct that substorms are phenomena in the magnetotail (exclusively).

............ Reply: Now we have corrected the misleading sentence in the revised manuscript as "Substorms are dynamic and transient phenomena that play a crucial role in the Earth's magnetosphere, storing solar wind energy and then releasing it through an explosive process"

100     .............

In addition to addressing the reviewer's comments, we have made further edits to the manuscript in response to the first reviewer, with all changes highlighted in blue.